# Iterative Deep Graph Learning for Graph Neural Networks: Better and Robust Node Embeddings

**Yu Chen**
Rensselaer Polytechnic Institute
cheny39@rpi.edu

**Lingfei Wu**\*
IBM Research
lwu@email.wm.edu

**Mohammed J. Zaki**
Rensselaer Polytechnic Institute
zaki@cs.rpi.edu

## Abstract

In this paper, we propose an end-to-end graph learning framework, namely **I**terative **D**eep **G**raph **L**earning (IDGL), for jointly and iteratively learning graph structure and graph embedding. The key rationale of IDGL is to learn a better graph structure based on better node embeddings, and vice versa (i.e., better node embeddings based on a better graph structure). Our iterative method dynamically stops when the learned graph structure approaches close enough to the graph optimized for the downstream prediction task. In addition, we cast the graph learning problem as a similarity metric learning problem and leverage adaptive graph regularization for controlling the quality of the learned graph. Finally, combining the anchor-based approximation technique, we further propose a scalable version of IDGL, namely IDGL-ANCH, which significantly reduces the time and space complexity of IDGL without compromising the performance. Our extensive experiments on nine benchmarks show that our proposed IDGL models can consistently outperform or match the state-of-the-art baselines. Furthermore, IDGL can be more robust to adversarial graphs and cope with both transductive and inductive learning.

## 1 Introduction

Recent years have seen a significantly growing amount of interest in graph neural networks (GNNs), especially on efforts devoted to developing more effective GNNs for node classification [29, 36, 17, 52], graph classification [60, 43] and graph generation [47, 37, 61]. Despite GNNs' powerful ability in learning expressive node embeddings, unfortunately, they can only be used when graph-structured data is available. Many real-world applications naturally admit network-structured data (e.g., social networks). However, these intrinsic graph-structures are not always optimal for the downstream tasks. This is partially because the raw graphs were constructed from the original feature space, which may not reflect the "true" graph topology after feature extraction and transformation. Another potential reason is that real-world graphs are often noisy or even incomplete due to the inevitably error-prone data measurement or collection. Furthermore, many applications such as those in natural language processing [7, 57, 58] may only have sequential data or even just the original feature matrix, requiring additional graph construction from the original data matrix.

To address these limitations, we propose an end-to-end graph learning framework, namely **I**terative **D**eep **G**raph **L**earning (IDGL), for jointly and iteratively learning the graph structure and the GNN parameters that are optimized toward the downstream prediction task. The key rationale of our IDGL

---

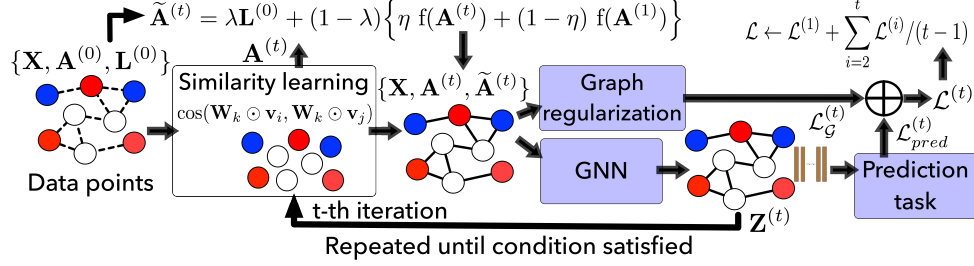

Figure 1: Overall architecture of the proposed IDGL framework. Dashed lines (in data points on left) indicate the initial noisy graph topology $\mathbf{A}^{(0)}$ (if not available we use a kNN graph).

framework is to learn a better graph structure based on better node embeddings, and at the same time, to learn better node embeddings based on a better graph structure. In particular, IDGL is a novel iterative method that aims to search for an implicit graph structure that augments the initial graph structure (if not available we use a kNN graph) with the goal of optimizing the graph for downstream prediction tasks. The iterative method adjusts when to stop in each mini-batch when the learned graph structure approaches close enough to the graph optimized for the downstream task.

Furthermore, we present a graph learning neural network that uses multi-head self-attention with epsilon-neighborhood sparsification for constructing a graph. Moreover, unlike the work in [25] that directly optimizes an adjacency matrix without considering the downstream task, we learn a graph metric learning function by optimizing a joint loss combining both task-specific prediction loss and graph regularization loss. Finally, we further propose a scalable version of our IDGL framework, namely IDGL-ANCH, by combining the anchor-based approximation technique, which reduces the time and memory complexity from quadratic to linear with respect to the numbers of graph nodes.

In short, we summarize the main contributions as follows:

- We propose a novel end-to-end graph learning framework (IDGL) for jointly and iteratively learning the graph structure and graph embedding. IDGL dynamically stops when the learned graph structure approaches the optimized graph (for prediction). To the best of our knowledge, we are the first to introduce iterative learning for graph structure learning.
- Combining the anchor-based approximation technique, we further propose a scalable version of IDGL, namely IDGL-ANCH, which achieves linear complexity in both computational time and memory consumption with respect to the number of graph nodes.
- Experimental results show that our models consistently outperform or match the state-of-the-art baselines on various downstream tasks. More importantly, IDGL can be more robust to adversarial graph examples and can cope with both transductive and inductive learning.

## 2 Iterative Deep Graph Learning Framework

### 2.1 Problem Formulation

Let the graph $\mathcal{G} = (\mathcal{V}, \mathcal{E})$ be represented as a set of $n$ nodes $v_i \in \mathcal{V}$ with an initial node feature matrix $\mathbf{X} \in \mathbb{R}^{d \times n}$, edges $(v_i, v_j) \in \mathcal{E}$ (binary or weighted) formulating an initial noisy adjacency matrix $\mathbf{A}^{(0)} \in \mathbb{R}^{n \times n}$, and a degree matrix $\mathbf{D}_{ii}^{(0)} = \sum_j \mathbf{A}_{ij}^{(0)}$. Given a noisy graph input $\mathcal{G} := \{\mathbf{A}^{(0)}, \mathbf{X}\}$ or only a feature matrix $\mathbf{X} \in \mathbb{R}^{d \times n}$, the deep graph learning problem we consider in this paper is to produce an optimized graph $\mathcal{G}^* := \{\mathbf{A}^{(*)}, \mathbf{X}\}$ and its corresponding graph node embeddings $\mathbf{Z} = f(\mathcal{G}^*, \theta) \in \mathbb{R}^{h \times n}$, with respect to some (semi-)supervised downstream task. It is worth noting that we assume that the graph noise is only from graph topology (the adjacency matrix) and the node feature matrix $\mathbf{X}$ is noiseless. The more challenging scenario where both graph topology and node feature matrix are noisy, is part of our future work. Without losing the generality, in this paper, we consider both node-level and graph-level prediction tasks.

### 2.2 Graph Learning and Graph Embedding: A Unified Perspective

Graph topology is crucial for a GNN to learn expressive graph node embeddings. Most of existing GNN methods simply assume that the input graph topology is perfect, which is not necessarily true

in practice since real-world graphs are often noisy or incomplete. More importantly, the provided input graph(s) may not be ideal for the supervised downstream tasks since most of raw graphs are constructed from the original feature space which may fail to reflect the "true" graph topology after high-level feature transformations. Some previous works [52] mitigate this issue by reweighting the importance of neighborhood node embeddings using self-attention on previously learned node embeddings, which still assumes that the original graph connectivity information is noiseless.

To handle potentially noisy input graph, we propose our novel IDGL framework that formulates the problem as an iterative learning problem which jointly learns the graph structure and the GNN parameters. The key rationale of our IDGL framework is to learn a better graph structure based on better node embeddings, and in the meanwhile, to learn better node embeddings based on a better graph structure, as shown in Fig. 2. Unlike most existing methods that construct graphs based on raw node features, the node embeddings learned by GNNs (optimized toward the downstream task) could provide useful information for learning better graph structures. On the other hand, the newly learned graph structures could be a better graph input for GNNs to learn better node embeddings.

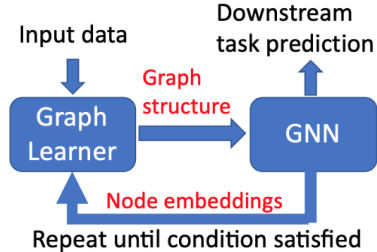

Figure 2: A sketch of the proposed IDGL framework.

In particular, IDGL is a novel iterative method that aims to search for an implicit graph structure that augments the initial graph structure (if not available we use a kNN graph) for downstream prediction tasks. The iterative method dynamically stops in each mini-batch when the learned graph structure approaches close enough to the optimized graph (with respect to the downstream task) based on our proposed stopping criterion. Moreover, the process of graph construction can be optimized toward the downstream task in an end-to-end manner.

## 2.3 Graph Learning as Similarity Metric Learning

Previous methods (e.g., [15]) that model the graph learning problem as learning a joint discrete probability distribution on the edges of the graph have shown promising performance. However, since they optimize the edge connectivities by assuming that the graph nodes are known, they are unable to cope with the inductive setting (with new nodes during testing). To overcome this issue, we cast the graph structure learning problem as similarity metric learning, which will be jointly trained with the prediction model dedicated to a downstream task.

**Graph similarity metric learning.** Common options for metric learning include cosine similarity [44, 54], radial basis function (RBF) kernel [59, 34] and attention mechanisms [51, 23]. A good similarity metric function is supposed to be learnable and expressively powerful. Although our framework is agnostic to various similarity metric functions, without loss of generality, we design a weighted cosine similarity as our metric function, $s_{ij} = \cos(\mathbf{w} \odot \mathbf{v}_i, \mathbf{w} \odot \mathbf{v}_j)$, where $\odot$ denotes the Hadamard product, and $\mathbf{w}$ is a learnable weight vector which has the same dimension as the input vectors $\mathbf{v}_i$ and $\mathbf{v}_j$, and learns to highlight different dimensions of the vectors. Note that the two input vectors could be either raw node features or computed node embeddings.

To stabilize the learning process and increase the expressive power, we extend our similarity metric function to a multi-head version (similar to the observations in [51, 52]). Specifically, we use $m$ weight vectors (each one representing one perspective) to compute $m$ independent similarity matrices using the above similarity function and take their average as the final similarity:

$$s_{ij}^p = \cos(\mathbf{w}_p \odot \mathbf{v}_i, \mathbf{w}_p \odot \mathbf{v}_j), \quad s_{ij} = \frac{1}{m} \sum_{p=1}^{m} s_{ij}^p \tag{1}$$

Intuitively, $s_{ij}^p$ computes the cosine similarity between the two input vectors $\mathbf{v}_i$ and $\mathbf{v}_j$, for the $p$-th perspective, where each perspective considers one part of the semantics captured in the vectors.

**Graph sparsification via $\varepsilon$-neighborhood.** Typically an adjacency matrix (computed from a metric) is supposed to be non-negative but $s_{ij}$ ranges between $[-1, 1]$. In addition, many underlying graph structures are much more sparse than a fully connected graph which is not only computationally expensive but also might introduce noise (i.e., unimportant edges). We hence proceed to extract a symmetric sparse non-negative adjacency matrix $\mathbf{A}$ from $\mathbf{S}$ by considering only the $\varepsilon$-neighborhood

for each node. Specifically, we mask off (i.e., set to zero) those elements in $\mathbf{S}$ which are smaller than a non-negative threshold $\varepsilon$.

**Anchor-based scalable metric learning.** The above similarity metric function like Eq. (1) computes similarity scores for all pairs of graph nodes, which requires $\mathcal{O}(n^2)$ complexity for both computational time and memory consumption, rendering significant scalablity issue for large graphs. To address the scalability issue, inspired by previous anchor-based methods [41, 55], we design an anchor-based scalable metric learning technique which learns a node-anchor affinity matrix $\mathbf{R} \in \mathbb{R}^{n \times s}$ (i.e., requires $\mathcal{O}(ns)$ for both time and space complexity where $s$ is the number of anchors) between the node set $\mathcal{V}$ and the anchor set $\mathcal{U}$. Note that $s$ is a hyperparameter which is tuned on the development set.

Specifically, we randomly sample a set of $s \in \mathcal{U}$ anchors from the node set $\mathcal{V}$, where $s$ is usually much smaller than $n$ in large graphs. The anchor embeddings are thus set to the corresponding node embeddings. Therefore, Eq. (1) can be rewritten as the following:

$$a_{ik}^p = \cos(\mathbf{w}_p \odot \mathbf{v}_i, \mathbf{w}_p \odot \mathbf{u}_k), \quad a_{ik} = \frac{1}{m} \sum_{p=1}^{m} a_{ik}^p \tag{2}$$

where $a_{ik}$ is the affinity score between node $v_i$ and anchor $u_k$. Similarly, we apply the $\varepsilon$-neighborhood sparsification technique to the node-anchor affinity scores $a_{ik}$ to obtain a sparse and non-negative node-anchor affinity matrix $\mathbf{R}$.

## 2.4 Graph Node Embeddings and Prediction

Although the initial graph could be noisy, it typically still carries rich and useful information regarding true graph topology. Ideally, the learned graph structure $\mathbf{A}$ could be supplementary to the original graph topology $\mathbf{A}^{(0)}$ to formulate an optimized graph for GNNs with respect to the downstream task. Therefore, with the mild assumption that the optimized graph structure is potentially a "shift" from the initial graph structure, we combine the learned graph with the initial graph,

$$\widetilde{\mathbf{A}}^{(t)} = \lambda \mathbf{L}^{(0)} + (1 - \lambda)\Big\{\eta\,\mathrm{f}(\mathbf{A}^{(t)}) + (1 - \eta)\,\mathrm{f}(\mathbf{A}^{(1)})\Big\} \tag{3}$$

where $\mathbf{L}^{(0)} = \mathbf{D}^{(0)^{-1/2}}\mathbf{A}^{(0)}\mathbf{D}^{(0)^{-1/2}}$ is the normalized adjacency matrix of the initial graph. $\mathbf{A}^{(t)}$ and $\mathbf{A}^{(1)}$ are the two adjacency matrices computed at the $t$-th and 1-st iterations (using Eq. (1)), respectively. The adjacency matrix is further row normalized, namely, $\mathrm{f}(\mathbf{A})_{ij} = A_{ij}/\sum_j A_{ij}$.

Note that $\mathbf{A}^{(1)}$ is computed from the raw node features $\mathbf{X}$, whereas $\mathbf{A}^{(t)}$ is computed from the previously updated node embeddings $\mathbf{Z}^{(t-1)}$ that is optimized toward the downstream prediction task. Therefore, we make the final learned graph structure as their linear combination weighted by a hyperparameter $\eta$, so as to combine the advantages of both. Finally, another hyperparameter $\lambda$ is used to balance the trade-off between the learned graph structure and the initial graph structure. If such an initial graph structure is not available, we instead use a kNN graph constructed based on raw node features $\mathbf{X}$ using cosine similarity.

Our graph learning framework is agnostic to various GNN architectures (that take as input a node feature matrix and an adjacency matrix to compute node embeddings) and prediction tasks. In this paper, we adopt a two-layered GCN [29] where the first layer (denoted as $\mathrm{GNN}_1$) maps the raw node features $\mathbf{X}$ to the intermediate embedding space, and the second layer (denoted as $\mathrm{GNN}_2$) further maps the intermediate node embeddings $\mathbf{Z}$ to the output space.

$$\mathbf{Z} = \mathrm{ReLU}(\mathrm{MP}(\mathbf{X}, \widetilde{\mathbf{A}})\mathbf{W}_1), \;\; \hat{\mathbf{y}} = \sigma(\mathrm{MP}(\mathbf{Z}, \widetilde{\mathbf{A}})\mathbf{W}_2), \;\; \mathcal{L}_{\mathrm{pred}} = \ell(\hat{\mathbf{y}}, \mathbf{y}) \tag{4}$$

where $\sigma(\cdot)$ and $\ell(\cdot)$ are task-dependent output function and loss function, respectively. For instance, for a classification task, $\sigma(\cdot)$ is a softmax function for predicting a probability distribution over a set of classes, and $\ell(\cdot)$ is a cross-entropy function for computing the prediction loss. $\mathrm{MP}(\cdot, \cdot)$ is a message passing function, and in GCN, $\mathrm{MP}(\mathbf{F}, \widetilde{\mathbf{A}}) = \widetilde{\mathbf{A}}\mathbf{F}$ for a feature/embedding matrix $\mathbf{F}$ and normalized adjacency matrix $\widetilde{\mathbf{A}}$ which we obtain using Eq. (3).

**Node-anchor message passing.** Note that a node-anchor affinity matrix $\mathbf{R}$ serves as a weighted adjacency matrix of a bipartite graph $\mathcal{B}$ allowing only direct connections between nodes and anchors. If we regard a direct travel between a node and an anchor as one-step transition described by $\mathbf{R}$, built

upon theories of stationary Markov random walks [42], we can actually recover both the node graph $\mathcal{G}$ and the anchor graph $\mathcal{Q}$ from $\mathbf{R}$ by computing the two-step transition probabilities. Let $\mathbf{A} \in \mathbb{R}^{n \times n}$ denote a row-normalized adjacency matrix for the node graph $\mathcal{G}$, and $A_{ij} = p^{(2)}(v_j|v_i)$ indicate the two-step transition probability from node $v_i$ to node $v_j$, $\mathbf{A}$ can be recovered from $\mathbf{R}$,

$$\mathbf{A} = \boldsymbol{\Delta}^{-1}\mathbf{R}\boldsymbol{\Lambda}^{-1}\mathbf{R}^{\top} \tag{5}$$

where $\Lambda \in \mathbb{R}^{s \times s}$ ($\Lambda_{kk} = \sum_{i=1}^{n} R_{ik}$) and $\Delta \in \mathbb{R}^{n \times n}$ ($\Delta_{ii} = \sum_{k=1}^{s} R_{ik}$) are both diagonal matrices. Similarly, we can recover the row-normalized adjacency matrix $\mathbf{B} \in \mathbb{R}^{s \times s}$ for the anchor graph $\mathcal{Q}$,

$$\mathbf{B} = \boldsymbol{\Lambda}^{-1}\mathbf{R}^{\top}\boldsymbol{\Delta}^{-1}\mathbf{R} \tag{6}$$

A detailed proof of recovering node and anchor graphs from the affinity matrix is provided in Appendix A.1. While explicitly computing a node adjacency matrix $\mathbf{A}$ from $\mathbf{R}$ (Eq. (5)) and directly performing message passing over the node graph $\mathcal{G}$ (Eq. (4)) are expensive in both time complexity ($\mathcal{O}(n^2 s)$) and space complexity ($\mathcal{O}(n^2)$), one can instead equivalently decompose the above process (denoted as $\text{MP}_{12}$) into two steps: i) node-to-anchor message passing $\text{MP}_1$ and ii) anchor-to-node message passing $\text{MP}_2$, over the node-anchor bipartite graph $\mathcal{B}$, formulated as follows,

$$\text{MP}_{12}(\mathbf{F}, \mathbf{R}) = \text{MP}_2(\mathbf{F}', \mathbf{R}), \quad \mathbf{F}' = \text{MP}_1(\mathbf{F}, \mathbf{R}) \tag{7}$$

where $\text{MP}_1(\mathbf{F}, \mathbf{R}) = \boldsymbol{\Lambda}^{-1}\mathbf{R}^{\top}\mathbf{F}$ aims to pass message $\mathbf{F}$ from the nodes $\mathcal{V}$ to the anchors $\mathcal{U}$, and $\text{MP}_2(\mathbf{F}', \mathbf{R}) = \boldsymbol{\Delta}^{-1}\mathbf{R}\mathbf{F}'$ aims to further pass the message $\mathbf{F}'$ aggregated on the anchors back to the nodes. Finally, we can obtain $\text{MP}_{12}(\mathbf{F}, \mathbf{R}) = \boldsymbol{\Delta}^{-1}\mathbf{R}\boldsymbol{\Lambda}^{-1}\mathbf{R}^{\top}\mathbf{F} = \mathbf{A}\mathbf{F}$ where $\mathbf{A}$ is the node adjacency matrix recovered from $\mathbf{R}$ using Eq. (5). In this way, we reduce both time and space complexity to $\mathcal{O}(ns)$. Therefore, we can rewrite the regular node embedding and prediction equations defined in Eqs. (3) and (4) as follows,

$$\mathbf{Z} = \text{ReLU}(\text{MP}_a(\mathbf{X}, \{\mathbf{L}^{(0)}, \mathbf{R}^{(t)}, \mathbf{R}^{(1)}\})\mathbf{W}_1), \quad \widehat{\mathbf{y}} = \sigma(\text{MP}_a(\mathbf{Z}, \{\mathbf{L}^{(0)}, \mathbf{R}^{(t)}, \mathbf{R}^{(1)}\})\mathbf{W}_2) \tag{8}$$

where $\text{MP}_a(\cdot, \cdot)$ is a hybrid message passing function with the same spirit of Eq. (3), defined as,

$$\text{MP}_a(\mathbf{F}, \{\mathbf{L}^{(0)}, \mathbf{R}^{(t)}, \mathbf{R}^{(1)}\}) = \lambda\text{MP}(\mathbf{F}, \mathbf{L}^{(0)}) + (1-\lambda)\Big\{\eta\text{MP}_{12}(\mathbf{F}, \mathbf{R}^{(t)}) + (1-\eta)\text{MP}_{12}(\mathbf{F}, \mathbf{R}^{(1)})\Big\} \tag{9}$$

Note that we use the same $\text{MP}(\cdot, \cdot)$ function defined in Eq. (4) for performing message passing over $\mathbf{L}^{(0)}$ which is typically sparse in practice, and $\mathbf{F}$ can either be $\mathbf{X}$ or $\mathbf{Z}$.

## 2.5  Graph Regularization

Although combining the learned graph $\mathbf{A}^{(t)}$ with the initial graph $\mathbf{A}^{(0)}$ is an effective way to approach the optimaized graph, the quality of the learned graph $\mathbf{A}^{(t)}$ plays an important role in improving the quality of the final graph $\widetilde{\mathbf{A}}^{(t)}$. In practice, it is important to control the smoothness, connectivity and sparsity of the resulting learned graph $\mathbf{A}^{(t)}$, which faithfully reflects the graph topology with respect to the initial node attributes $\mathbf{X}$ and the downstream task.

Let each column of the feature matrix $\mathbf{X}$ be considered as a graph signal. A widely adopted assumption for graph signals is that values change smoothly across adjacent nodes. Given an undirected graph with a symmetric weighted adjacency matrix $A$, the smoothness of a set of $n$ graph signals $\mathbf{x}_1, \ldots, \mathbf{x}_n \in \mathbb{R}^d$ is usually measured by the Dirichlet energy [2],

$$\Omega(\mathbf{A}, \mathbf{X}) = \frac{1}{2n^2}\sum_{i,j} A_{ij}||\mathbf{x}_i - \mathbf{x}_j||^2 = \frac{1}{n^2}\text{tr}(\mathbf{X}^T\mathbf{L}\mathbf{X}) \tag{10}$$

where $\text{tr}(\cdot)$ denotes the trace of a matrix, $\mathbf{L} = \mathbf{D} - \mathbf{A}$ is the graph Laplacian, and $\mathbf{D} = \sum_j A_{ij}$ is the degree matrix. As can be seen, minimizing $\Omega(\mathbf{A}, \mathbf{X})$ forces adjacent nodes to have similar features, thus enforcing smoothness of the graph signals on the graph associated with $\mathbf{A}$.

However, solely minimizing the smoothness loss will result in the trivial solution $\mathbf{A} = 0$. Also, it is desirable to have control of how sparse the resulting graph is. Following [25], we impose additional constraints on the learned graph,

$$f(\mathbf{A}) = \frac{-\beta}{n}\mathbf{1}^T\log(\mathbf{A1}) + \frac{\gamma}{n^2}||\mathbf{A}||_F^2 \tag{11}$$

---

**Algorithm 1** General Framework for IDGL and IDGL-ANCH

---
1: **Input:** $\mathbf{X}, \mathbf{y}[, \mathbf{A}^{(0)}]$
2: **Parameters:** $m, \varepsilon, \alpha, \beta, \gamma, \lambda, \delta, T, \eta, k[, s]$
3: **Output:** $\Theta, \widehat{\mathbf{y}}, \widetilde{\mathbf{A}}^{(t)}$ or $\mathbf{R}^{(t)}$
4: $[\mathbf{A}^{(0)} \leftarrow \text{kNN}(\mathbf{X}, k)]$ {kNN-graph if no initial $\mathbf{A}^{(0)}$}
5: $t \leftarrow 1$
6: StopCond $\leftarrow |\mathbf{A}^{(t)} - \mathbf{A}^{(t-1)}|_F^2 > \delta|\mathbf{A}^{(1)}|_F^2$ **if** IDGL **else** $|\mathbf{R}^{(t)} - \mathbf{R}^{(t-1)}|_F^2 > \delta|\mathbf{R}^{(t)}|_F^2$
7: **while** $((t == 1 \text{ or StopCond}) \text{ and } t \leqslant T$ **do**
8:     **if** IDGL **then**
9:         $\mathbf{A}^{(t)} \leftarrow \text{GL}(\mathbf{X})$ **or** $\text{GL}(\mathbf{Z}^{(t-1)})$ using Eq. (1) {Refine adj. matrix}
10:         $\widetilde{\mathbf{A}}^{(t)} \leftarrow \{\mathbf{A}^{(0)}, \mathbf{A}^{(t)}, \mathbf{A}^{(1)}\}$ using Eq. (3) {Combine refined and raw adj. matrices}
11:         $\mathbf{Z}^{(t)} \leftarrow \text{GNN}_1(\widetilde{\mathbf{A}}^{(t)}, \mathbf{X})$ using Eq. (4) {Refine node embeddings}
12:     **else**
13:         $\mathbf{R}^{(t)} \leftarrow \text{GL}(\mathbf{X}, \mathbf{X}_{\mathcal{U}})$ **or** $\text{GL}(\mathbf{Z}^{(t-1)}, \mathbf{Z}_{\mathcal{U}}^{(t-1)})$ using Eq. (2) {Refine affinity matrix}
14:         $\mathbf{Z}^{(t)} \leftarrow \text{GNN}_1(\{\mathbf{A}^{(0)}, \mathbf{R}^{(t)}, \mathbf{R}^{(1)}\}, \mathbf{X})$ using Eqs. (8) and (9) {Refine node embeddings}
15:     **end if**
16:     $\widehat{\mathbf{y}} \leftarrow \text{GNN}_2(\widetilde{\mathbf{A}}^{(t)}, \mathbf{Z}^{(t)})$ using Eq. (4) **if** IDGL **else** $\text{GNN}_2(\{\mathbf{A}^{(0)}, \mathbf{R}^{(t)}, \mathbf{R}^{(1)}\}, \mathbf{Z}^{(t)})$ using Eqs. (8) and (9)
17:     $\mathcal{L}_{\text{pred}}^{(t)} \leftarrow \text{LOSS}_1(\widehat{\mathbf{y}}, \mathbf{y})$ using Eq. (4)
18:     $\mathcal{L}_{\mathcal{G}}^{(t)} \leftarrow \alpha\Omega(\mathbf{A}^{(t)}, \mathbf{X}) + f(\mathbf{A}^{(t)})$ **if** IDGL **else** $\alpha\Omega(\widehat{\mathbf{B}}^{(t)}, \mathbf{X}^{\mathcal{U}}) + f(\widehat{\mathbf{B}}^{(t)})$ **where** $\widehat{\mathbf{B}}^{(t)} = \mathbf{R}^{(t)\top}\mathbf{\Delta}^{-1}\mathbf{R}^{(t)}$
19:     $\mathcal{L}^{(t)} \leftarrow \mathcal{L}_{\text{pred}}^{(t)} + \mathcal{L}_{\mathcal{G}}^{(t)}$ **and** $t \leftarrow t + 1$
20: **end while**
21: $\mathcal{L} \leftarrow \mathcal{L}^{(1)} + \sum_{i=2}^{t}\mathcal{L}^{(i)}/(t-1)$
22: Back-propagate $\mathcal{L}$ to update model weights $\Theta$ {In training phase only}

---

where $|| \cdot ||_F$ denotes the Frobenius norm of a matrix. The first term penalizes the formation of disconnected graphs via the logarithmic barrier, and the second term controls sparsity by penalizing large degrees due to the first term.

We then define the overall graph regularization loss as the sum of the above losses $\mathcal{L}_{\mathcal{G}} = \alpha\Omega(\mathbf{A}, \mathbf{X}) + f(\mathbf{A})$, which is able to control the smoothness, connectivity and sparsity of the learned graph where $\alpha, \beta$ and $\gamma$ are all non-negative hyperparameters.

**Anchor graph regularization.** As shown in Eq. (6), we can obtain a row-normalized adjacency matrix $\mathbf{B}$ for the anchor graph $\mathcal{Q}$ in $\mathcal{O}(ns^2)$ time complexity. In order to control the quality of the learned node-anchor affinity matrix $\mathbf{R}$ (which can result in implicit control of the quality of the node adjacency matrix $\mathbf{A}$), we apply the aforementioned graph regularization techniques to the anchor graph. It is worthing noting that our proposed graph regularization loss is only applicable to non-negative and symmetric adjacency matrices [26]. Therefore, instead of applying graph regularization to $\mathbf{B}$ which is often not symmetric, we opt to apply graph regularization to its unnormalized version $\widehat{\mathbf{B}} = \mathbf{R}^{\top}\mathbf{\Delta}^{-1}\mathbf{R}$ as $\mathcal{L}_{\mathcal{G}} = \alpha\Omega(\widehat{\mathbf{B}}, \mathbf{X}^{\mathcal{U}}) + f(\widehat{\mathbf{B}})$, where $\mathbf{X}^{\mathcal{U}}$ denotes the set of anchor embeddings sampled from the set of node embeddings $\mathbf{X}$.

## 2.6 Joint Learning with A Hybrid Loss

Compared to previous works which directly optimize the adjacency matrix based on either graph regularization loss [26], or task-dependent prediction loss [15], we propose to jointly and iteratively learning the graph structure and the GNN parameters by minimizing a hybrid loss function combining both the task prediction loss and the graph regularization loss, namely, $\mathcal{L} = \mathcal{L}_{\text{pred}} + \mathcal{L}_{\mathcal{G}}$.

The full algorithm of the IDGL framework is presented in Algorithm 1. As we can see, our model repeatedly refines the adjacency matrix with updated node embeddings (Eq. (1)), and refines the node embeddings (Eqs. (3) and (4)) with the updated adjacency matrix until the difference between adjacency matrices at consecutive iterations are smaller than certain threshold. Note that compared to using a fixed number of iterations globally, our dynamic stopping criterion is more beneficial, especially for mini-batch training. At each iteration, a hybrid loss combining both the task-dependent prediction loss and the graph regularization loss is computed. After all iterations, the overall loss is back-propagated through all previous iterations to update the model parameters. Notably, Algorithm 1

is also applicable to IDGL-ANCH. The major differences between IDGL and IDGL-ANCH are how we compute adjacency (or affinity) matrix, and perform message passing and graph regularization.

## 3 Experiments

In this section, we conduct extensive experiments to verify the effectiveness of IDGL and IDGL-ANCH in various settings. The implementation of our proposed models is publicly available at `https://github.com/hugochan/IDGL`.

**Datasets and baselines.** The benchmarks used in our experiments include four citation network datasets (i.e., Cora, Citeseer, Pubmed and ogbn-arxiv) [48, 21] where the graph topology is available, three non-graph datasets (i.e., Wine, Breast Cancer (Cancer) and Digits) [11] where the graph topology does not exist, and two text benchmarks (i.e., 20Newsgroups data (20News) and movie review data (MRD)) [32, 46] where we treat a document as a graph containing each word as a node. The first seven datasets are all for node classification tasks in the transductive setting, and we follow the experimental setup of previous works [29, 15, 21]. The later two datasets are for graph-level prediction tasks in the inductive setting. Please refer to Appendix C.1 for detailed data statistics.

Our main baseline is LDS [15] which however is incapable of handling inductive learning problems, we hence only report its results on transductive datasets. In addition, for citation network datasets, we include other GNN variants (i.e., GCN [29], GAT [52], GraphSAGE [18], APPNP [30], H-GCN [20] and GDC [31]) as baselines. For non-graph and text benchmarks where the graph topology is unavailable, we conceive various GNN$_{kNN}$ baselines (i.e., GCN$_{kNN}$, GAT$_{kNN}$ and GraphSAGE$_{kNN}$) where a kNN graph on the data set is constructed during preprocessing before applying a GNN model. For text benchmarks, we include a BiLSTM [19] baseline. The reported results are averaged over 5 runs with different random seeds.

**Experimental results.** Table 1 shows the results of transductive experiments. First of all, IDGL outperforms all baselines in 4 out of 5 benchmarks. Besides, compared to IDGL, IDGL-ANCH is more scalable and can achieve comparable or even better results. In a scenario where the graph structure is available, compared to the state-of-the-art GNNs and graph learning models, our models achieve either significantly better or competitive results, even though the underlying GNN component of our models is a vanilla GCN. When the graph topology is not available (thus GNNs are not directly applicable), compared to graph learning baselines, IDGL consistently achieves much better results on all datasets. Compared to our main graph learning baseline LDS, our models not only achieve significantly better performance, but also are more scalable. The results of inductive experiments are shown in Table 2. Unlike LDS which cannot handle inductive setting, the good performance on 20News and MRD demonstrates the capability of IDGL on inductive learning.

Table 1: Summary of results in terms of classification accuracies (in percent) on transductive benchmarks. The star symbol indicates that we ran the experiments. The dash symbol indicates that reported results were unavailable or we were not able to run the experiments due to memory issue.

| Model | Cora | Citeseer | Pubmed | ogbn-arxiv | Wine | Cancer | Digits |
|---|---|---|---|---|---|---|---|
| GCN | 81.5 | 70.3 | 79.0 | 71.7 (0.3) | — | — | — |
| GAT | 83.0 (0.7) | 72.5 (0.7) | 79.0 (0.3) | — | — | — | — |
| GraphSAGE | 77.4 (1.0) | 67.0 (1.0) | 76.6 (0.8) | 71.5 (0.3) | — | — | — |
| APPNP | — | **75.7 (0.3)** | 79.7 (0.3) | — | — | — | — |
| H-GCN | **84.5 (0.5)** | 72.8 (0.5) | 79.8 (0.4) | — | — | — | — |
| GCN+GDC | 83.6 (0.2) | 73.4 (0.3) | 78.7 (0.4) | — | — | — | — |
| LDS | 84.1 (0.4) | 75.0 (0.4) | — | — | 97.3 (0.4) | 94.4 (1.9) | 92.5 (0.7) |
| GCN$_{kNN}$* | — | — | — | — | 95.9 (0.9) | 94.7 (1.2) | 89.5 (1.3) |
| GAT$_{kNN}$* | — | — | — | — | 95.8 (3.1) | 88.6 (2.7) | 89.8 (0.6) |
| GraphSAGE$_{kNN}$* | — | — | — | — | 96.5 (1.1) | 92.8 (1.0) | 88.4 (1.8) |
| LDS* | 83.9 (0.6) | 74.8 (0.3) | — | — | 96.9 (1.4) | 93.4 (2.4) | 90.8 (2.5) |
| IDGL | **84.5 (0.3)** | 74.1 (0.2) | — | — | 97.8 (0.6) | **95.1 (1.0)** | 93.1 (0.5) |
| IDGL-ANCH | 84.4 (0.2) | 72.0 (1.0) | **83.0 (0.2)** | **72.0 (0.3)** | **98.1 (1.1)** | 94.8 (1.4) | **93.2 (0.9)** |

**Ablation study.** Table 3 shows the ablation study results on different modules in our models. we can see a significant performance drop consistently for both IDGL and IDGL-ANCH on all datasets by turning off the iterative learning component (i.e., iterating only once), indicating its effectiveness. Besides, we can see the benefits of jointly training the model with the graph regularization loss.

Table 2: Summary of results in terms of classification accuracies or regression scores ($R^2$) (in percent) on inductive benchmarks.

| Methods | 20News | MRD |
|---|---|---|
| BiLSTM | 80.0 (0.4) | 53.1 (1.4) |
| GCN$_{kNN}$ | 81.3 (0.6) | 60.1 (1.5) |
| IDGL | **83.6 (0.4)** | **63.7 (1.8)** |
| IDGL-Anch | 82.9 (0.3) | 62.9 (0.4) |

Table 3: Ablation study on various node/graph classification datasets.

| Methods | Cora | Citeseer | Wine | Cancer | Digits | 20News |
|---|---|---|---|---|---|---|
| IDGL | **84.5 (0.3)** | **74.1 (0.2)** | **97.8 (0.6)** | **95.1 (1.0)** | **93.1 (0.5)** | **83.6 (0.4)** |
| w/o graph reg. | 84.3 (0.4) | 71.5 (0.9) | 97.3 (0.8) | 94.9 (1.0) | 91.5 (0.9) | 83.4 (0.5) |
| w/o IL | 83.5 (0.6) | 71.0 (0.8) | 97.2 (0.8) | 94.7 (0.9) | 92.4 (0.4) | 83.0 (0.4) |
| IDGL-Anch | **84.4 (0.2)** | **72.0 (1.0)** | **98.1 (1.1)** | **94.8 (1.4)** | **93.2 (0.9)** | **82.9 (0.3)** |
| w/o graph reg. | 83.2 (0.8) | 70.1 (0.8) | 97.4 (1.8) | **94.8 (1.4)** | 92.0 (1.3) | 82.5 (0.7) |
| w/o IL | 83.6 (0.2) | 68.6 (0.7) | 96.4 (1.5) | 94.0 (2.6) | 93.0 (0.4) | 82.3 (0.3) |

**Model analysis.** To evaluate the robustness of IDGL to adversarial graphs, we construct graphs with random edge deletions or additions. Specifically, for each pair of nodes in the original graph, we randomly remove (if an edge exists) or add (if no such edge) an edge with a probability 25%, 50% or 75%. As shown in Fig. 3, compared to GCN and LDS, IDGL achieves better or comparable results in both scenarios. While both GCN and LDS completely fail in the edge addition scenario, IDGL performs reasonably well. We conjecture this is because the edge addition scenario is more challenging than the edge deletion scenario by incorporating misleading additive random noise to the initial graph. And Eq. (3) is formulated as a form of skip-connection, by lowering the value of $\lambda$ (i.e., tuned on the development set), we enforce the model to rely less on the initial noisy graph.

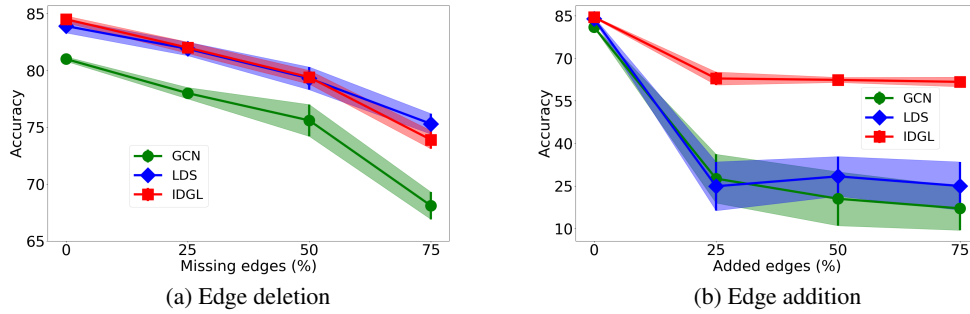

(a) Edge deletion

(b) Edge addition

Figure 3: Test accuracy ($\pm$ standard deviation) in percent for the edge attack scenarios on Cora.

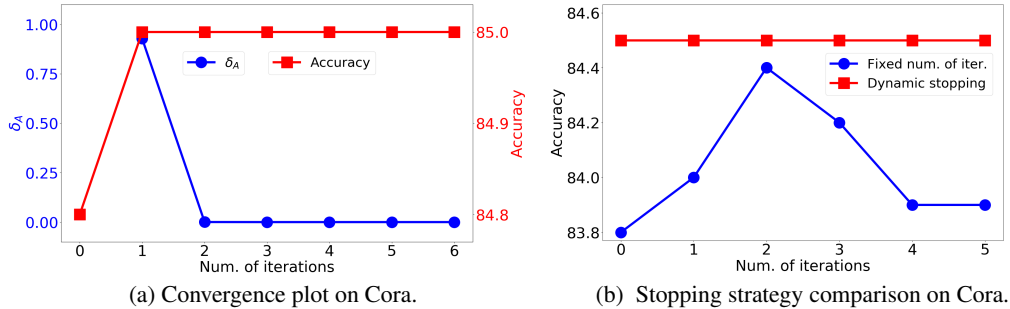

(a) Convergence plot on Cora.

(b) Stopping strategy comparison on Cora.

Figure 4: Convergence and stopping strategy study on Cora (Single run results).

In Fig. 4a (and Appendix B.1), we show the evolution of the learned adjacency matrix and accuracy through iterations in the iterative learning procedure in the testing phase. We compute the difference between adjacency matrices at consecutive iterations as $\delta_A^{(t)} = ||\mathbf{A}^{(t)} - \mathbf{A}^{(t-1)}||_F^2 / ||\mathbf{A}^{(t)}||_F^2$ which typically ranges from 0 to 1. As we can see, both the adjacency matrix and accuracy converge quickly. This empirically verifies the analysis we made on the convergence property of IDGL in Appendix A.2. Please note that this convergence property is not due to the oversmoothing effect of GNNs [56, 33], because we only employ a two-layered GCN as the underlying GNN module of IDGL in our experiments.

We compare the training efficiency of IDGL and IDGL-ANCH with other baselines. As shown in Table 4, IDGL is consistently faster than LDS, but in general, they are comparable. Note that IDGL has comparable model size compared to LDS. For instance, on the Cora data, the number of trainable parameters of IDGL is 28,836, and for LDS, it is 23,040. And we see a large speedup of IDGL-ANCH compared to IDGL. Note that we were not able to run IDGL on Pubmed because of memory limitation. The theoretical complexity analysis is provided in Appendix A.3.

We also empirically study the stopping strategy (Fig. 4b and Appendix B.2), visualize the graph structures learned by IDGL (Appendix B.3), and conduct hyperparameter analysis (Appendix B.4). Details on model settings are provided in Appendix C.2.

## 4 Related Work

The problem of graph structure learning has been widely studied in different fields from different perspectives. In the field of graph signal processing, researchers have explored various ways of learning graphs from data [10, 12, 53, 27, 3, 1], with certain structural constraints (e.g., sparsity) on the graphs. This problem has also been studied in the literature of clustering analysis [4, 22] where they aimed to simultaneously perform the clustering task and learn similarity relationships among objects. These works all focused on unsupervised learning setting

Table 4: Mean and standard deviation of training time (5 runs) on various benchmarks (in seconds).

| Data | Cora | Citeseer | Pubmed |
|---|---|---|---|
| GCN | **3 (1)** | **5 (1)** | **29 (4)** |
| GAT | 26 (5) | 28 (5) | — |
| LDS | 390 (82) | 585 (181) | — |
| IDGL | 237 (21) | 563 (100) | — |
| w/o IL | 49 (8) | 61 (15) | — |
| IDGL-ANCH | 83 (6) | 261 (50) | 323 (53) |
| w/o IL | 28 (4) | 69 (9) | 71 (17) |

without considering any supervised downstream tasks, and were incapable of handling inductive learning problems. Other related works include structure inference in probabilistic graphical models [9, 66, 62], and graph generation [38, 49], which have a different goal from ours.

In the field of GNNs [29, 16, 18, 35, 63], there is a line of research on developing robust GNNs [50] that are invulnerable to adversarial graphs by leveraging attention-based methods [5], Bayesian methods [13, 64], graph diffusion-based methods [31], and various assumptions on graphs (e.g., low rank and sparsity) [14, 24, 65]. These methods usually assume that the initial graph structure is available. Recently, researchers have explored methods to automatically construct a graph of objects [45, 8, 34, 15, 40] or words [39, 6, 7] when applying GNNs to non-graph structured data. However, these methods merely optimize the graphs toward the downstream tasks without the explicit control of the quality of the learned graphs. More recently, [15] proposed the LDS model for jointly learning the graph and the parameters of GNNs by leveraging the bilevel optimization technique. However, by design, their method is unable to handle the inductive setting. Our work is also related to Transformer-like approaches [51] that learn relationships among objects by leveraging multi-head attention mechanism. However, these methods do not focus on the graph learning problem and were not designed to utilize the initial graph structure.

## 5 Conclusion

We proposed a novel IDGL framework for jointly and iteratively learning the graph structure and embeddings optimized for the downstream task. Experimental results demonstrate the effectiveness and efficiency of the proposed models. In the future, we plan to explore effective techniques for handling more challenging scenarios where both graph topology and node features are noisy.

## Broader Impact

The fundamental goal of our research is to develop a method for jointly learning graph structures and embeddings that are optimized for (semi-)supervised downstream tasks. Our technique can be widely applied to a large range of applications, including social network analysis, natural language processing (e.g., question answering and text generation), drug discovery and community detection. Conceptually, any application with the purpose of jointly learning the graph structures and embeddings in order to perform well in downstream tasks. Those potential applications range from computer vision, natural language processing, and network analysis. For instance, our research might be used to help better capture the semantic relationships between word tokens (beyond a sequence of tokens) in natural language processing.

There are many benefits of using our method as a tool, such as applying graph neural networks to non-graph structured data without manual graph construction, and learning node/graph embeddings that are more robust to noisy input graphs. Those benefits that might be utilized by a large number of potential applications may have a board range of societal impacts:

- the use of our research could improve and speed up the process of learning meaningful graphs from noisy/incomplete graphs (e.g., social networks) or even non-graph structured data (e.g., text and images).
- the use of our research could improve the robustness of graph neural networks to noisy/incomplete graph-structured data in terms of learning good node/graph embeddings for downstream task.

We would encourage the research to explore similar approaches in more specific real-world applications. We would also suggest the research to understand the adversarial robustness of the use of graph neural networks in safety/security-critical applications.

## Acknowledgments and Disclosure of Funding

This research is sponsored by the Defense Advanced Research Projects Agency (DARPA) through Cooperative Agreement D20AC00004 awarded by the U.S. Department of the Interior (DOI), Interior Business Center. The content of the information does not necessarily reflect the position or the policy of the Government, and no official endorsement should be inferred. This work is also supported by IBM Research AI through the IBM AI Horizons Network. We thank the anonymous reviewers for their constructive suggestions.

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
