[Supplementary Material]

## A  Theoretical Model Analysis

### A.1  Theoretical Proof of Recovering Node and Anchor Graphs from Affinity Matrix R

It is worth noting that a node-anchor affinity matrix $\mathbf{R}$ serves as a weighted adjacency matrix of a bipartite graph $\mathcal{B}$. We hence establish stationary Markov random walks [40] by defining the one-step transition probabilities as follows,

$$p^{(1)}(u_k|v_i) = \frac{R_{ik}}{\sum_{k'=1}^s R_{ik'}}, \quad p^{(1)}(v_i|u_k) = \frac{R_{ik}}{\sum_{i'=1}^n R_{i'k}}, \quad \forall v_i \in \mathcal{V}, \quad \forall u_k \in \mathcal{U} \quad (12)$$

We can further compute the two-step transition probabilities between nodes as follows,

$$p^{(2)}(v_j|v_i) = \sum_{k=1}^s p^{(1)}(v_j|u_k)p^{(1)}(u_k|v_i) = \sum_{k=1}^s \frac{R_{jk}}{\sum_{j'=1}^n R_{j'k}} \frac{R_{ik}}{\sum_{k'=1}^s R_{ik'}} = \sum_{k=1}^s \frac{R_{jk}}{\Lambda_{kk}} \frac{R_{ik}}{\Delta_{ii}} \quad (13)$$

where $\Lambda_{kk} = \sum_{j'=1}^n R_{j'k}$ and $\Delta_{ii} = \sum_{k'=1}^s R_{ik'}$. Therefore, we can recover a row-normalized adjacency matrix $\mathbf{A} \in \mathbb{R}^{n \times n}$ for the node graph as $A_{ij} = p^{(2)}(v_j|v_i)$, which can be further written in a compact form $\mathbf{A} = \mathbf{\Delta}^{-1}\mathbf{R}\mathbf{\Lambda}^{-1}\mathbf{R}^\top$.

Similarly, we can compute the two-step transition probabilities between anchors as follows,

$$p^{(2)}(u_r|u_k) = \sum_{i=1}^n p^{(1)}(u_r|v_i)p^{(1)}(v_i|u_k) = \sum_{i=1}^n \frac{R_{ir}}{\sum_{r'=1}^s R_{ir'}} \frac{R_{ik}}{\sum_{i'=1}^n R_{i'k}} = \sum_{i=1}^n \frac{R_{ir}}{\Delta_{ii}} \frac{R_{ik}}{\Lambda_{kk}} \quad (14)$$

And a row-normalized adjacency matrix $\mathbf{B} \in \mathbb{R}^{s \times s}$ for the anchor graph $\mathcal{Q}$ can be formulated as $B_{kr} = p^{(2)}(u_r|u_k)$. And we can obtain $\mathbf{B} = \mathbf{\Lambda}^{-1}\mathbf{R}^\top \mathbf{\Delta}^{-1}\mathbf{R}$.

### A.2  Theoretical Convergence Analysis

While it is challenging to theoretically prove the convergence of the proposed iterative learning procedure due to the arbitrary complexity of the model, here we want to conceptually understand why it works in practice. Fig. 5 shows the information flow of the learned adjacency matrix $\mathbf{A}$ and the updated node embedding matrix $\mathbf{Z}$ during the iterative procedure. For the sake of simplicity, we omit some other variables such as $\widetilde{\mathbf{A}}$. As we can see, at $t$-th iteration, $\mathbf{A}^{(t)}$ is computed based on $\mathbf{Z}^{(t-1)}$ (Line 9), and $\mathbf{Z}^{(t)}$ is computed based on $\widetilde{\mathbf{A}}^{(t)}$ (Line 11) which is computed based on $\mathbf{A}^{(t)}$ (Eq. (3)). We further denote the difference between the adjacency matrices at the $t$-th iteration and the previous iteration by $\delta_A^{(t)}$. Similarly, we denote the difference between the node embedding matrices at the $t$-th iteration and the previous iteration by $\delta_Z^{(t)}$.

Figure 5: Information flow of iterative learning procedure.

If we assume that $\delta_Z^{(2)} < \delta_Z^{(1)}$, then we can expect that $\delta_A^{(3)} < \delta_A^{(2)}$ because conceptually a more similar node embedding matrix (i.e., smaller $\delta_Z$) is supposed to produce a more similar adjacency matrix (i.e., smaller $\delta_A$) given the fact that model parameters keep the same through iterations. Similarly, given that $\delta_A^{(3)} < \delta_A^{(2)}$, we can expect that $\delta_Z^{(3)} < \delta_Z^{(2)}$. Following this chain of reasoning, we can easily extend it to later iterations. In order to see why the assumption $\delta_Z^{(2)} < \delta_Z^{(1)}$ makes sense in practice, we need to recall the fact that $\delta_Z^{(1)}$ measures the difference between $\mathbf{Z}^{(1)}$ and $\mathbf{X}$, which is

usually larger than the difference between $\mathbf{Z}^{(2)}$ and $\mathbf{Z}^{(1)}$, namely $\delta_Z^{(2)}$. For example, the raw node feature matrix $\mathbf{X}$ can be quite sparse in practice (e.g., in Cora and Citeseer), whereas $\mathbf{Z}^{(1)}$ is typically a dense matrix.

### A.3 Model Complexity Analysis

As for IDGL, the cost of learning an adjacency matrix is $\mathcal{O}(n^2 h)$ for $n$ nodes and data in $\mathbb{R}^h$, while computing node embeddings costs $\mathcal{O}(n^2 h + ndh)$, computing task output costs $\mathcal{O}(n^2 d)$, and computing the total loss costs $\mathcal{O}(n^2 d)$ where $d$ is the hidden size. We set the maximal number of iterations to $T$, hence the overall complexity is $\mathcal{O}(Tn(nh + nd + hd))$. If we assume that $d \approx h$ and $n \gg d$, the overall time complexity is $\mathcal{O}(Tdn^2)$.

As for IDGL-ANCH, the cost of learning a node-anchor affinity matrix is $\mathcal{O}(nsh)$, while computing node embeddings costs $\mathcal{O}(nsh + ndh + |\mathcal{E}|h)$, computing task output costs $\mathcal{O}(nsd + |\mathcal{E}|d)$, and computing the total loss costs $\mathcal{O}(ns^2 + s^2 d)$ where $|\mathcal{E}|$ is the number of edges in the initial or kNN graph $\mathcal{G}$. With the assumption that the initial or kNN graph is usually very sparse in practice, especially for large graphs, we hence set $|\mathcal{E}| = kn$ where $k$ is a constant denoting the average degree of the initial or kNN graph. Therefore, we get the overall time complexity $\mathcal{O}(Tn(ds + d^2 + s^2))$. If we assume that $n \gg s$ which usually holds true for large graphs, the overall time complexity is linear with respect to the numbers of graph nodes $n$.

As for space complexity, compared to IDGL, IDGL-ANCH reduces it from $\mathcal{O}(n^2)$ to $\mathcal{O}(ns)$ since it only needs to store the $n \times s$ affinity matrix.

## B   Empirical Model Analysis

### B.1   Convergence Test

Here, we show the evolution of the learned adjacency matrix and accuracy through iterations in the iterative learning procedure in the testing phase. As we can see, both the adjacency matrix and accuracy converge quickly.

Figure 6: Convergence study on Cora (single run results).

Figure 7: Convergence study on Citeseer (single run results).

## B.2 Stopping Strategy Analysis

Here, we empirically compare the effectiveness of two stopping strategies: i) using a fixed number of iterations (blue line), and ii) using a stopping criterion to dynamically determine the convergence (red line). As we can see, dynamically adjusting the number of iterations using the stopping criterion works better in practice. Compared to using a fixed number of iterations globally, the advantage of applying this dynamical stopping strategy becomes more clear when we are doing mini-batch training since we can adjust when to stop dynamically for each example graph in the mini-batch.

Figure 8: Performance comparison (i.e., test accuracy in %) of two different stopping strategies on Cora.

Figure 9: Performance comparison (i.e., test accuracy in %) of two different stopping strategies on Citeseer.

## B.3 Graph Visualization

Here, we visualize the graph structures (i.e., $\mathbf{A}^{(t)}$) learned by IDGL. As we can see, compared to the initial graph structures, IDGL mainly forms graph structures within the same class of nodes, which complement the initial graph structure. This is as expected because $\mathbf{A}^{(t)}$ is computed based on the updated node embeddings that are supposed to capture certain node label information.

## B.4 Hyperparameter Analysis

A hyperparameter $\lambda$ is used to balance the trade-off between using the learned graph structure and the initial (or kNN) graph structure. In Table 5, we show the results of using different values of $\lambda$ on Cora.

We also study the effect of the hyperparameter $s$ (i.e., the number of anchors in IDGL-ANCH). As shown in Table 6, lower value of $s$ can degrade the performance of IDGL-ANCH whereas after certain optimal value, further increasing the number of anchors might not help the performance.

(a) Initial graph ($\mathbf{A}^{(0)}$)  (b) Learned graph ($\mathbf{A}^{(t)}$)

Figure 10: Visualization of the initial graph and the learned graph on Cora. Colors indicate different node labels.

(a) kNN graph ($\mathbf{A}^{(0)}$)  (b) Learned graph ($\mathbf{A}^{(t)}$)

Figure 11: Visualization of the kNN graph and the learned graph on Wine. Colors indicate different node labels.

Table 5: Test scores ($\pm$ standard deviation) with different values of $\lambda$ on the Cora data.

| Methods / $\lambda$ | 0.9 | 0.8 | 0.7 | 0.6 | 0.5 |
|---|---|---|---|---|---|
| IDGL | 83.6 (0.4) | **84.5 (0.3)** | 83.9 (0.3) | 82.4 (0.1) | 80.9 (0.2) |
| IDGL-ANCH | 83.2 (0.4) | **84.4 (0.2)** | 83.5 (0.6) | 82.9 (0.4) | 54.6 (32.3) |

Table 6: Test scores ($\pm$ standard deviation) with different values of $s$ for IDGL-ANCH on the Cora and Pubmed data.

| Methods / $s$ | 1,600 | 1,300 | 1,000 | 700 | 400 | 100 |
|---|---|---|---|---|---|---|
| Cora | 84.0 (0.4) | 84.1 (0.5) | **84.4 (0.2)** | 83.8 (0.2) | 58.7 (30.5) | 38.3 (25.9) |
| Pubmed | 82.7 (0.2) | **83.0 (0.4)** | 82.7 (0.4) | **83.0 (0.2)** | 82.7 (0.3) | 82.4 (0.5) |

## C Details on Experimental Setup

### C.1 Data Statistics

Table 7 shows the data statistics of the nine benchmarks used in our experiments.

Table 7: Data statistics. (clf. indicates classification and reg. indicates regression.)

| Benchmarks | #Nodes | #Edges | Train/Dev/Test | Task | Setting |
|---|---|---|---|---|---|
| Cora | 2,708 (1 graph) | 5,429 | 140/500/1,000 | node clf. | transductive |
| Citeseer | 3,327 (1 graph) | 4,732 | 120/500/1,000 | node clf. | transductive |
| Pubmed | 19,717 (1 graph) | 44,338 | 60/500/1,000 | node clf. | transductive |
| ogbn-arxiv | 169,343 (1 graph) | 1,166,243 | 90,941/29,799/48,603 | node clf. | transductive |
| Wine | 178 (1 graph) | N/A | 10/20/158 | node clf. | transductive |
| Cancer | 569 (1 graph) | N/A | 10/20/539 | node clf. | transductive |
| Digits | 1,797 (1 graph) | N/A | 50/100/1,647 | node clf. | transductive |
| 20News | 317 (18,846 graphs) | N/A | 7,919/3,395/7,532 | graph clf. | inductive |
| MRD | 389 (5,006 graphs) | N/A | 3,003/1,001/1,002 | graph reg. | inductive |

### C.2 Model Settings

Table 8: Hyperparameter for IDGL on all benchmarks.

| Benchmarks | $\lambda$ | $\eta$ | $\alpha$ | $\beta$ | $\gamma$ | $k$ | $\epsilon$ | $m$ | $\delta$ | $T$ |
|---|---|---|---|---|---|---|---|---|---|---|
| Cora | 0.8 | 0.1 | 0.2 | 0.0 | 0.0 | – | 0.0 | 4 | 4.0e-5 | 10 |
| Citeseer | 0.6 | 0.5 | 0.4 | 0.0 | 0.2 | – | 0.3 | 1.0 | 1.0e-3 | 10 |
| Wine | 0.8 | 0.7 | 0.1 | 0.1 | 0.3 | 20 | 0.75 | 1 | 1.0e-3 | 10 |
| Cancer | 0.25 | 0.1 | 0.4 | 0.2 | 0.1 | 40 | 0.9 | 1 | 1.0e-3 | 10 |
| Digits | 0.4 | 0.1 | 0.4 | 0.1 | 0.0 | 24 | 0.65 | 8 | 1.0e-4 | 10 |
| 20News | 0.1 | 0.4 | 0.5 | 0.01 | 0.3 | 950 | 0.3 | 12 | 8.0e-3 | 10 |
| MRD | 0.5 | 0.9 | 0.2 | 0.0 | 0.1 | 350 | 0.4 | 5 | 4.0e-2 | 10 |

Table 9: Hyperparameter for IDGL-ANCH on all benchmarks.

| Benchmarks | $\lambda$ | $\eta$ | $\alpha$ | $\beta$ | $\gamma$ | $k$ | $\epsilon$ | $m$ | $\delta$ | $T$ | num./ratio of anchors |
|---|---|---|---|---|---|---|---|---|---|---|---|
| Cora | 0.8 | 0.1 | 0.2 | 0.0 | 0.1 | – | 0.0 | 4 | 8.5e-5 | 10 | 1,000 |
| Citeseer | 0.6 | 0.5 | 0.5 | 0.1 | 0.2 | – | 0.2 | 4 | 2.0e-3 | 10 | 1,400 |
| Pubmed | 0.7 | 0.3 | 0.0 | 0.03 | 0.0 | – | 0.1 | 6 | 8.0e-5 | 10 | 700 |
| ogbn-arxiv | 0.8 | 0.1 | 0.2 | 0.0 | 0.0 | – | 0.9 | 1 | 1.0e-1 | 2 | 300 |
| Wine | 0.7 | 0.7 | 0.1 | 0.1 | 0.3 | 20 | 0.75 | 1 | 1.0e-3 | 10 | 200 |
| Cancer | 0.25 | 0.1 | 0.0 | 0.0 | 0.0 | 40 | 0.9 | 4 | 8.0e-4 | 10 | 100 |
| Digits | 0.3 | 0.3 | 0.4 | 0.1 | 0.0 | 24 | 0.65 | 8 | 1.0e-4 | 10 | 1,500 |
| 20News | 0.1 | 0.3 | 0.4 | 0.0 | 0.3 | 950 | 0.4 | 12 | 1.0e-2 | 10 | 0.4 |
| MRD | 0.5 | 0.75 | 0.2 | 0.0 | 0.0 | 400 | 0.7 | 4 | 3.0e-2 | 10 | 0.4 |

In all our experiments, we apply a dropout ratio of 0.5 after GCN layers except for the output GCN layer. During the iterative learning procedure, we also apply a dropout ratio of 0.5 after the intermediate GCN layer, except for Citeseer (no dropout) and Digits (0.3 dropout). For experiments on text benchmarks, we keep and fix the 300-dim GloVe vectors for words that appear more than 10 times in the dataset. For long documents, for the sake of efficiency, we cut the text length to maximum 1,000 words. We apply a dropout ratio of 0.5 after word embedding layers and BiLSTM layers. The batch size is set to 16. And the hidden size is set to 128 and 64 for 20News and MRD, respectively. For all other benchmarks, the hidden size is set to 16 to follow the original GCN paper. For the text benchmarks, we apply a BiLSTM to a sequence of word embeddings. The concatenation of the last forward and backward hidden states of the BiLSTM is used as the initial node features. We use Adam [26] as the optimizer. For the text benchmarks, we set the learning rate to 1e-3. For all other benchmarks, we set the learning rate to 0.01 and apply L2 norm regularization with weight decay set to 5e-4. As for IDGL-ANCH, we set the number of anchors as a hyperparameter in transductive experiments, while in inductive experiments, we set the ratio of anchors (proportional to the graph

size) as a hyperparameter. In Table 8 and Table 9, we show the hyperparameters for IDGL and IDGL-Anch on all benchmarks, respectively. All hyperparameters are tuned on the development set.