[Reviews · NeurIPS 2020]

Review 1

Summary and Contributions: This manuscript proposes a new deep graph learning framework which integrates feature learning and graph structure learning for a link prediction task. Based on anchor-based metric learning, a scalable method is also proposed to accelerate the computation of the original model. The proposed method slightly outperforms other state-of-the-art models.

Strengths: 1. The manuscript is readable. 2. The idea of alternatively optimizing the node embedding vectors and revising the graph structure seems to be reasonable even though the effect of this integration is somewhat limitedly observed in the experimental results. 3. The anchor-based metric learning effectively reduces the model complexity while not losing much accuracy.

Weaknesses: The main algorithm (Algorithm 1) should be described in the main text instead of Appendix. I was able to understand the overall flow of the algorithm only after reading Algorithm 1 in Appendix. In Table 1, it is kind of disappointing to see that the performance gap between the proposed method and the best baseline method is so little on all the datasets. Since the proposed method is a more expensive method than other methods as shown in Table 4, one might expect to see this cost is payed off by a large performance increase. However, I'm afraid that Table 1 does not support this. Also, it would be better to show the performance of all the graph-based models (GCN, GAT, GraphSage, and so on) on the Wine, Cancer, Digits datasets by generating a simple kNN graph based on the features. (Currently, only GCN_{kNN} is presented.) It would be informative if we can check the performances of all the graph-based models in the kNN graph. In the experiments, it would be better to include more 'link prediction' tasks and include more datasets related to the link prediction task because the main goal of the proposed method is to increase the accuracy in link prediction.

Correctness: Overall, the reasoning of the main idea and the proposed algorithm seem to be reasonable even though I did not check all the details. The experiments use benchmark datasets and standard metrics to compare the performances of the methods.

Clarity: Algorithm 1 should be presented in the main text, or there should be some high-level description about the overall algorithm somewhere in the main text. In the experiments, how the training set and the test set is divided should be described in the manuscript.

Relation to Prior Work: It seems that the proposed method is properly compared with other related methods.

Reproducibility: No

Additional Feedback:


Review 2

Summary and Contributions: In this paper, the author presented a new graph learning method for graph neural networks. The authors started to analyze the significant drawbacks of existing GNNs methods: 1. work only when the graph data input is given; 2. ignore potentially imperfect graph inputs (due to the noise and cannot reflect true graph topology); 3. completely fail when inputs like texts are not given in graph format. To solve these problems, this paper proposed a new deep graph learning framework for learning the graph embedding and graph structure at the same time. Specifically, this paper introduced an iterative deep graph learning approach, where the key idea is to alternatively produce a better and more robust graph node embedding with a better learned graph structure and then learn a better graph structure based on better graph node embeddings. They further proposed a scalable version of the proposed method IDGL by leveraging the anchor-based approximation method. Graph similarity learning and graph regularization are also proposed to learn a graph structure with controlled quality, instead of learning a fully connected graph in existing methods. Authors show the experimental comparisons with other state-of-the-art GNNs methods. The proposed method showed better performance on accuracy and computational time. In addition, the presented method showed good scalability even on relatively large dataset without compromising the accuracy, owning to the use of anchor-based approximation technique.

Strengths: (1) The idea of using iterative method for constructing a graph and learning graph node embeddings in a joint fashion seems very novel, to the best of my knowledge. Authors further presented improved version of IDGL in order to overcome the scalability issue of original IDGL for dealing with large graphs. Authors also provided theoretical analysis of the convergence of Anchor graphs to the exact similarity graph, making the proposed method more principled. (2) Instead of using fully connected graph (e.g. employing self-attention/transformer-like techniques), authors tend to use sparsified graph over the fully connected graph by performing epsilon-neighborhood and adaptive graph regularization and showed better performance in the experiments. I found this insight is very interesting and could be transferred to other domain as well. (3) The paper is very well written and clearly motivated. For instance, the paper started to discuss the issues of existing methods (assuming graph is perfect, LDS only dealt with transductive setting and has scalability issue) and they proposed their solutions. These issues are well explained in the paper and the authors do a good job at explaining how their proposed method addresses these issues. (4) The overall experimental and ablation study results well investigated several advantages of the method in terms of both accuracy and scalability compared to existing GNNs methods.

Weaknesses: (1) Authors discussed anchor-based metric learning and showed how it can use affinity matrix between graphs nodes and anchor nodes to reduce both runtimes and memory consumption. I think this is very interesting direction for dealing with large-scale graphs. However, I did not see any experiments how the convergence of performance when increasing the number of anchor points. More detailed discussion on the sensitivity of the number of anchor nodes should be given. (2) Authors did not provide the full algorithm 1 in the main section, which I found it is very inconvenient to digest the whole framework. I would suggest authors to move it in the main section in order to better discuss them in details. (3) It seems like this work is also related to adversarial robustness of graph neural networks. I would like to see more discussions regarding the connection and difference between this work and adversarial examples in GNN domain.

Correctness: Yes

Clarity: Yes

Relation to Prior Work: Yes

Reproducibility: Yes

Additional Feedback:


Review 3

Summary and Contributions: This paper focus on the problem of how to develop a (scalable) graph learning technique, which has been underexplored in the domain. Existing popular methods still have some limitations: first, GNNs can be only used when graph-structured data is provided; second, the intrinsic graph inputs are often noisy (not optimal) or incomplete; third, many applications may even only have sequence data or data matrix and lack of graph structured data. To overcome such issues, this paper presented a novel end-to-end graph learning framework, called Iterative Deep Graph Learning (IDGL), to joint learn graph structure and graph embeddings. The model can generate more robust and better node embedding when learning graph structure through an iterative learning framework. It also has lower computational complexity (IDGL-ANCH) with the anchor-based approximation. The model can cope with both transductive and inductive settings, where the learned embeddings can be applied for many tasks. Besides, the key ideas of the paper, i.e., iterative graph learning and anchor approximation, seem generally useful for other machine learning tasks and problems. The author also performed extensive set of experiments to study the properties of the proposed methods and demonstrated the effectiveness and efficiency compared to other popular GNN and graph learning methods.

Strengths: (1) This paper studies an important problem and I like the overall idea behind the work. A simple and elegant solution is proposed to develop robust and better node embeddings, and at the same time, learn graph structure using a novel iterative approach. (2) The model has several key components including graph similarity learning, graph regularization, joint learning, and particular anchor-based graph approximation. They are designed and combined in a nice architecture for fulfilling the above mentioned properties, although some of them are not completely new in the vast of literature. (3) A good set of baselines and datasets are chosen in the experiments and an appropriate set of experiments are performed to show the strong empirical performance of the IDGL and IDGL-ANCH in comparison to other state-of-the-art baselines. In fact, the ablation study and model analyses help me better understand the roles of each important component, which brings me good points.

Weaknesses: (1) In lines 139-143, the authors mentioned the final graph is the combination of the initial graph topology and learned graph topology. However, it is unclear to me why we also have the combination of A(t) and A(1) in Eq. (3). I suggest that elaborating more on this part. (2) It seems like the graph embedding component is fixed to GCN in the paper. But it seems to me that it can be any existing variant of GNNs. Could you elaborate more on it? Also, have you tried to use different variants of GNNs to combine with your IDGL framework? I suggest adding some analyses about this point. (3) Minors: Some hyper parameters need better explanations (such as lamda, gamma, beta and eta) and the paper needs some editing to fix a few grammar issues.

Correctness: Yes, The model is in a nice architecture and the experiments have shown the effectiveness of each part.

Clarity: Yes, the paper is well written, which is easy to follow, although some hyper parameters needs better explanations.

Relation to Prior Work: Yes, The paper has pointed out the differences between the proposed model and existing works and highlighted the contributions.

Reproducibility: Yes

Additional Feedback: Thanks for the authors' response. I hope my suggestions would be helpful.


Review 4

Summary and Contributions: Many real-world applications naturally admit network-structured data (e.g., social networks). However, these intrinsic graph-structures are not always optimal for the downstream tasks. To address this real-world issue, this paper presents an end-to-end graph learning framework, namely Iterative Deep Graph Learning (IDGL), for jointly and iteratively learning the graph structure and the GNN parameters that are optimized towards the downstream prediction task. The performance of the proposed IDGL is consistently better than or close to that of SOTA baselines. The ablation study is provided to verify each component in IDGL is effective. In general, this paper is a good paper and should be presented in NeurIPS2020.

Strengths: This paper addresses an important problem existed in GNNs. When training a GNN, many real-world applications naturally admit network-structured data (e.g., social networks). However, these intrinsic graph-structures are not always optimal for the downstream tasks. To address this real-world issue, this paper presents an end-to-end graph learning framework, namely Iterative Deep Graph Learning (IDGL), for jointly and iteratively learning the graph structure and the GNN parameters that are optimized towards the downstream prediction task. The performance of the proposed IDGL is consistently better than or close to that of SOTA baselines. The ablation study is provided to verify each component in IDGL is effective.

Weaknesses: I would be the best to provide an empirical evidence that intrinsic graph-structures are not always optimal for the downstream tasks in the introduction section.

Correctness: Yes, all claims and methods are correct. The the empirical methodology is correct.

Clarity: This paper is well written.

Relation to Prior Work: Yes. it is clearly discussed how this work differs from previous contributions.

Reproducibility: Yes

Additional Feedback: Many real-world applications naturally admit network-structured data (e.g., social networks). However, these intrinsic graph-structures are not always optimal for the downstream tasks. To address this real-world issue, this paper presents an end-to-end graph learning framework, namely Iterative Deep Graph Learning (IDGL), for jointly and iteratively learning the graph structure and the GNN parameters that are optimized towards the downstream prediction task. The performance of the proposed IDGL is consistently better than or close to that of SOTA baselines. The ablation study is provided to verify each component in IDGL is effective. In general, this paper is a good paper and should be presented in NeurIPS2020. The detailed comments can be seen below. 1. It would be the best to provide empirical evidence that intrinsic graph-structures are not always optimal for the downstream tasks (in the introduction section). 2. Line 35 says that "if not available we use a kNN graph". How does the k value affect the performance of the proposed method? 3. The motivation behind the graph regularization is unclear. Some references should be added to support using this regularizer. 4. Since the second contribution is related to time complexity, the theoretical time complexity analysis (and the running time) should be presented in the main contents rather than in the appendix. 5. Similar to the 4th comment, the whole algorithm should be moved to the main content.

[Author Response · NeurIPS 2020]

We thank all reviewers for their thorough reading and valuable comments! Please find below our responses.

**Response to Review #2:**

**On the downstream tasks:** It seems like there is a misunderstanding on the downstream tasks we considered in this
paper. We do not dedicate our approach to the link prediction task even though our approach aims to jointly learn graph
structures and node embeddings (that are optimized towards the downstream node/graph classification tasks). In some
scenarios where the initial graph structure (could be noisy or incomplete) is provided, our goal is to learn "optimized"
graph structures and node embeddings for the downstream task. In addition, in many other scenarios, no ground-truth
graph structure is available and thus neither linkage information at hand. Therefore, in our experiments, we choose
node-level and graph-level prediction tasks such as node classification, graph classification and graph regression.

**On the significance of experimental results:** our baselines include both state-of-the-art GNN models (which can
only be used when graph structure is available) and graph learning models. In scenarios where the graph structure is
available, compared to the state-of-the-art GNNs, our proposed models achieve either significantly better (e.g., 2.9%
absolute performance gain on Pubmed data) or competitive results, even though the underlying GNN component of our
models is just a vanilla GCN. In scenarios where the graph structure is not available (thus regular GNNs are not directly
applicable), compared to graph learning baselines, our proposed models consistently achieve much better results on
almost all datasets. Compared to our main graph learning baseline LDS, our models not only achieve significantly
better performance, but also are more scalable (i.e., our IDGL-ANCH model). More importantly, our IDGL models
are able to handle both transductive and inductive problems (whereas LDS can only handle transductive problems).

**On the effectiveness of iterative learning:** as shown in Table 2, the performance gain of utilizing our iterative
learning strategy is actually significant and consistent by comparing IDGL vs. IDGL w/o IL, and IDGL-ANCH vs.
IDGL-ANCH w/o IL on six datasets. In some datasets, the performance gap is even huge (e.g., more than 3% absolute
performance gain on Citeseer data).

**New results of other kNN-GNN variants on Wine, Cancer, Digits:** kNN-GAT: 95.8 (3.1) | 88.6 (2.7) | 89.8 (0.6),
kNN-GraphSAGE: 96.5 (1.1) | 92.8 (1.0) | 88.4 (1.8). For your reference, below are the reported results of kNN-
GCN: 95.9 (0.9) | 94.7 (1.2) | 89.5 (1.3) and IDGL: **97.8 (0.6)** | **95.1 (1.0)**| **93.1 (0.5)**. It is easy to see that our IDGL
consistently outperforms the state-of-the-art in kNN graph.

**On the training time:** as shown in Table 4, the training time of our proposed IDGL model is comparable to our main
graph learning baseline LDS. In addition, the training time of our IDGL-ANCH variant is much lower than LDS, and
is even comparable to the GAT model which is already fairly fast.

**On data splits:** we did mention the data split statistics in Line 242-244. As for the seven datasets in Table 1, we
followed the experimental setup of previous works. As for the two datasets in Table 3, due to the space limit, we put
the data split statistics in Appendix D.1. We will move the necessary data statistics to the main text in the revision.

**On reproducibility**: we described our experimental settings in Line 237-244 as well as our model settings in Appendix
D.2. In addition, we provided the source code and instructions as supplementary files for better reproducibility.

**Response to Review #3:**

**On the number of anchor nodes:** Thanks for the great suggestions. we reported the sensitivity of the number of
anchor nodes in Table 6 in the Appendix. We will add more discussions regarding that. We will report the effect of the
number of anchor nodes on the convergence in the revision.

**On adversarial GNNs:** we will add more discussions on the connections and difference between our work and
adversarial defenses in GNNs in the revision.

**Response to Review #4:**

**On combining A(1) and A(t):** Thanks for the great suggestions. As mentioned in Line 147-150, in order to combine
the advantages of both A(1) (computed from the raw node features) and A(t) (computed from the updated node
embeddings that are optimized toward the downstream task), we compute a weighted sum of them, as shown in
Eq. (3). We will add more discussions in the revision.

**On GNN components:** in principle, our proposed framework is agnostic to any GNN that takes as input node feature
matrix and adjacency matrix to compute node embeddings. We choose a vanilla GCN because it is simple and widely
used. It will be interesting to apply our framework to other GNN variants. We will discuss this point in the revision.

**On explanations of hyperparameters:** we will add more explanations on important hyperparameters.

**Final remark:** we hope our replies resolve all reviewers' concerns and are helpful in making the final recommenda-
tion.

[Meta-Review · NeurIPS 2020]

This paper focuses on the problem of how to develop a (scalable) graph learning technique, which has been underexplored in the domain. The proposal is a novel end-to-end graph learning framework to joint learn graph structure and graph embeddings. The philosophy behind sounds quite interesting to me, namely, sparsified graph over the fully connected graph by performing epsilon-neighborhood and adaptive graph regularization. This philosophy leads to a novel algorithm design I have never seen, i.e., Iterative Deep Graph Learning (IDGL). More importantly, IDGL can cope with both transductive and inductive settings, where the learned embeddings can be applied for many tasks. The clarity and novelty are clearly above the bar of NeurIPS. While the reviewers had some concerns on the significance, the authors did a particularly good job in their rebuttal. Thus, most of us have agreed to accept this paper for publication! Please carefully address R2' comments in the final version.